# The introduction of genetic counseling in Ethiopia: Results of a training workshop and lessons learned

Shane C. Quinonez[1,2]*, Bridget C. O'Connor[1], Michelle F. Jacobs[2], Atnafu Mekonnen Tekleab[3], Ayalew Marye[4], Delayehu Bekele[4], Beverly M. Yashar[5], Erika Hanson[2], Abate Yeshidinber[3], Getahun Wedaje[3]

1 Division of Pediatric Genetics, Metabolism and Genomic Medicine, Department of Pediatrics, Michigan Medicine, Ann Arbor, Michigan, United States of America, 2 Division of Genetic Medicine, Department of Internal Medicine, University of Michigan, Ann Arbor, Michigan, United States of America, 3 Department of Pediatrics, St. Paul's Hospital Millennium Medical College, Addis Ababa, Ethiopia, 4 Department of Obstetrics and Gynecology, Saint Paul's Hospital Millennium Medical College, Addis Ababa, Ethiopia, 5 Department of Human Genetics, University of Michigan, Ann Arbor, Michigan, United States of America

* squinon@med.umich.edu

**Data Availability Statement:** All relevant data are within the manuscript and its Supporting Information files.

## Abstract

### Background

Over the past two decades non-communicable diseases (NCDs) have steadily increased as a cause of worldwide disability and mortality with a concomitant decrease in disease burden from communicable, maternal, neonatal and nutritional conditions. Congenital anomalies, the most common NCD affecting children, have recently become the fifth leading cause of under-five mortality worldwide, ahead of other conditions such as malaria, neonatal sepsis and malnutrition. Genetic counseling has been shown to be an effective method to decrease the impact of congenital anomalies and genetic conditions but is absent in almost all sub-Saharan Africa countries. To address this need for counseling services we designed and implemented the first broad-based genetic counseling curriculum in Ethiopia, launching it at St. Paul's Millennium Medical College (SPHMMC) in Addis Ababa, Ethiopia.

### Methods

The curriculum, created by Michigan Medicine and SPHMMC specialists, consisted of medical knowledge and genetic counseling content and was delivered to two cohorts of nurses. Curriculum evaluation consisted of satisfaction surveys and pre- and post-assessments covering medical knowledge and genetic counseling content. Following Cohort 1 training, the curriculum was modified to increase the medical knowledge material and decrease Western genetic counseling principles material.

### Results

Both cohorts reported high levels of satisfaction but felt the workshop was too short. No significant improvements in assessment scores were seen for Cohort 1 in terms of total scores and medical knowledge and genetic counseling-specific questions. Following curriculum

**Funding:** SCQ, BCO and MFJ were previously supported by the Center for International Reproductive Health Training (CIRHT) for their work on this project https://cirht.med.umich.edu/. The funders had no role in study design, data collection and analysis, decision to publish, or preparation of the manuscript.

**Competing interests:** The authors have declared that no competing interests exist.

modification, improvements were seen in Cohort 2 with an increase in total assessment scores from 63% to 73% (p = 0.043), with medical knowledge-specific questions increasing from 57% to 79% (p = 0.01) with no significant change in genetic counseling-specific scores. Multiple logistic, financial, cultural and systems-specific barriers were identified with recommendations for their consideration presented.

## Conclusion

Genetics medical knowledge of Ethiopian nurses increased significantly following curriculum delivery though difficulty was encountered with Western genetic counseling material.

## Introduction

Every year 7.9 million children are born with a serious congenital anomaly of genetic or partially genetic origin [1]. While congenital anomalies are a global problem, low- and middle-income countries (LMICs) shoulder the major disease burden as 94 percent of births with serious congenital anomalies and 95 percent of the deaths from these conditions occur in LMICs [1]. Over the past two decades non-communicable diseases (NCDs) have steadily increased as a cause of worldwide disability and mortality with a concomitant decrease in disease burden from communicable, maternal, neonatal and nutritional conditions [2]. Congenital anomalies, the most common NCD affecting children, have recently become the fifth leading cause of under-five mortality worldwide, ahead of other conditions such as malaria, neonatal sepsis and malnutrition [3]. This epidemiologic transition, currently ongoing in most LMICs, was previously seen in many high-income countries in the 1960s and led to the well-established preventative, diagnostic and management medical genetics and public health services present today [4]. Unfortunately, many LMICs lack the necessary personnel, infrastructure and technology to implement identical approaches used in many high-income countries [5, 6]. To address this gap in services various strategies have been recommended all requiring careful consideration of a country's unique cultural, religious, legal and epidemiologic characteristics [4, 6].

Genetic counseling, the process of helping people understand and adapt to the medical, psychological and familial implications of genetic contributions to disease, has been shown to be an effective method to decrease the impact of congenital anomalies and genetic conditions [7–11]. Though genetic counseling is well-established in many higher-income countries, except for South Africa it is absent in almost all other sub-Saharan Africa (SSA) countries and many other LMICs [12]. Condition-specific genetic counseling has been reported in a few SSA countries including prenatal services in Cameroon and counseling as part of retinoblastoma care in Kenya [8, 10]. All countries will inevitably gain access to next-generation sequencing in some form in the future making it increasingly important to consider optimal ways of providing genetic counseling [7, 13, 14].

Ethiopia, a low-income country in Eastern Africa, has increasingly focused on NCDs including improved capacity to diagnose, prevent and treat congenital anomalies [15]. Currently there are no established medical genetics services with no in-country clinical geneticists or genetic counselors. Previous work though has shown Ethiopian patients are interested in diagnostic and preventative prenatal services with healthcare providers expressing a need for further training in medical genetics and genetic counseling [16, 17]. Limited clinical genetic testing consisting of multiplex ligation probe amplification is available through a private lab in

Ethiopia since 2014 with plans for platform expansion in the future to include next-generation sequencing. A medical genetics-needs assessment that has been ongoing at St. Paul's Hospital Millennium Medical College (SPHMMC), located in Addis Ababa, identified a need for further provider education in medical genetics. Providers highlighted the need for genetic counseling education to support established prenatal diagnostic services and pediatric providers caring for patients with congenital anomalies and genetic conditions such as clinically diagnosed Trisomy 21, seen frequently in the pediatrics ward [17]. Though genetic counseling is not an established practice in Ethiopia, patients at SPHMMC felt the services provided by a genetic counselor were in line with their values and indicated they would trust delivered information [18].

To address this need for counseling services we designed and implemented the first broad-based genetic counseling curriculum for providers at SPHMMC. The curriculum was designed in collaboration with Ethiopian physicians and delivered to two cohorts of SPHMMC nurses utilizing a Train the Trainer model. Here we report the results of the curriculum including satisfaction data as well as pre- and post-training knowledge assessments. We additionally provide information on lessons learned as there were several unanticipated barriers to the training and eventual implementation of the material. The information presented here will be useful to others planning similar trainings which are likely to increase in the future with the necessary expansion of medical genetics services in SSA and in other LMICs.

## Materials and methods

### Curriculum design

In response to expanding reproductive services at SPHMMC through the effort of the Center for International Reproductive Health Training (CIRHT) a study author (SQ) was asked to design and implement a targeted genetic counseling training program for SPHMMC obstetrical and pediatric nurses. A description of the practice setting has been previously reported [16, 17]. A workgroup was created consisting of 3 physicians (1 US-based geneticist and 2 Ethiopian physicians) and 3 US-based genetic counselors, one of whom has prior experience developing an international genetic counseling curriculum. A needs assessment was performed in May 2018 to identify Ethiopian public and provider interest in genetic counseling services, to determine the highest yield conditions to cover during the training and to address logistical aspects of the curriculum. Unstructured interviews of nurses were performed in the prenatal ultrasound clinic, neonatal intensive care unit (NICU), pediatrics ward and Michu clinic which provides counseling, family planning and safe abortion care. The interviews focused on how nurses were utilized in clinical spaces, interactions of nurses with physicians and patients and patients' preferences and ideal locations for genetic counseling services utilization. The Michu clinic, given its well established reproductive services counseling, quiet and private clinical setting, cohort of senior nurses and excellent lines of communication between nurses and physicians, was identified as the ideal location for future best practices determination.

A survey of patients showing positive views and preferences of genetic counseling services was conducted and has been published elsewhere [18]. The insights gained from this needs assessment were combined with previously established recommendations of core competences in genetics essential for all health professionals [19].

### Cohort training

The following lecture topics were finalized and delivered to Cohort 1 over 3 half-days in October 2018: DNA Basics and Genetic Diagnosis, Introduction to Genetic Disease, Introduction to Genetic Counseling, Genetic Disease (covering aneuploidies, congenital heart disease and

neural tube defects), Inheritance Patterns and Pedigree Generation, Psychosocial Counseling, Facilitated Decision-making, Delivering Bad News and Handling Patient Emotion. Cohort 1 consisted of 7 SPHMMC nurses (3 obstetrical nurses and 4 pediatric nurses) with content delivered via 9 lectures, 9 problem sets, 3 small group discussions and a role-playing exercise. Cohort 1 training occurred in a SPHMMC conference room. Since the Ethiopian medical curriculum is taught in English, the curriculum was delivered in English by three study authors (SQ, BO, MJ) who were aided by an Ethiopian physician who translated confusing topics to trainees and/or questions to the trainers in Amharic, the principal language in Ethiopia. While medical education in Ethiopia is generally taught in English, Amharic and Afan Oromo are the most commonly spoken languages at SPHMMC with patients. Attempts were made to choose nurses with adequate English fluency for ease of training. Curriculum evaluation utilized a novel 30 question pre- and post-assessment consisting of human genetics medical knowledge and genetic counseling principle questions and open response questions evaluating individual lectures and the entire curriculum (S1 Assessment). All evaluations were provided and completed in English. For Cohort 1 and Cohort 2 pre-assessments were completed on the first day of the workshop. Individual lecture evaluations were completed at the end of each day with the entire curriculum evaluation and post-assessment completed at the end of the final day of the workshop.

Based on appraisal of assessments, participant surveys and trainer observations curriculum modifications were implemented to increase the focus on human genetics medical knowledge and decrease the amount of didactic content covering Western-based genetic counseling principles, including facilitating decision-making, handling patient emotion and delivering bad news. It was felt these principles were the most difficult to grasp by nurses and were more likely to misalign with established Ethiopian cultural norms [18]. The modified curriculum was delivered over 4 half-days in July 2019 and covered the following topics: DNA Basics and Genetic Diagnosis, Introduction to Genetic Disease, Introduction to Genetic Counseling, Genetic Disease, Psychosocial Counseling, Pedigree Generation and Inheritance Patterns. The curriculum was delivered by one study author (SQ), 2 previously trained nurses from Cohort 1 and a translator. The lecture on psychosocial counseling was delivered entirely by a previously trained nurse from Cohort 1. Cohort 2 training occurred at a hotel located near SPHMMC given the number of participants increased to 15 with 13 nurses from the Obstetrics department and 2 from the Pediatrics department. The time allotted by the extra half day was used to review previously covered material in depth and allowed for the incorporation of more problem-based examples of delivered content. A total of 57 supplemental problem-based examples were utilized for Cohort 2 with the majority covering pedigree generation and inheritance patterns. This was a significant increase from Cohort 1 which utilized less than 10 supplemental problem-based examples. Problem-based examples were designed during the sessions utilizing a large-sized easel pad.

Trainees were provided with a genetic counseling checklist (S1 Checklist) to review the aspects of a genetic counseling visit but also as a quality assurance/improvement instrument. Additionally, case log sheets were provided so the curriculum workgroup could track how often nurses were providing counseling and the indications for counseling.

## Statistical analysis

Assessment questions were mapped to previously identified core competencies essential for all health professions [19]. For each competency, an average score was calculated based on the total number of correct answers to the questions that covered its content. For example, in Cohort 1 with seven participants, if a competency was only covered by 2 questions, the total

possible score was 14 given that a total number of 14 correct answers were possible (2 questions multiplied by 7 participants). The competencies most represented were competency 1.1 Basic genetics human terminology and 1.13 The components of the genetic counseling process and indications for referral to a genetic specialist. Assessment-represented competencies were divided into those covering genetics medical knowledge and those specifically covering genetic counseling. Though curriculum modifications were made, assessments and represented competencies were identical for both cohorts. **Table 1** shows assessed competencies and the number of assessment questions covering each competency.

Assessment data and survey responses were collected in Excel. R was used to calculate pooled Student's t-tests to assess statistically significant differences between pre- and post-assessments for each competency. For True/False assessment questions though individuals were able to provide an explanation for their True/False response this aspect of the question was not scored.

This work received Institutional Research Board approval from the SPHMMC Institutional Review Board. Informed consent was not obtained as SPHMMC does not require informed consent for educational research.

## Results

### Cohort results

For Cohort 1 pre-workshop assessment showed a total score of 63% (19/30) with an increase to 70% (21/30) following training though this change was not statistically significant (p = 0.07). The medical knowledge content specific scores increased in absolute score from 57% (8/14) to 64% (9/14) and again approached statistical significance but with a p-value of

**Table 1. Assessment genetics core competencies.**

| Medical Knowledge (number of questions*) | Genetic Counseling (number of questions*) |
|---|---|
| 1.1 Basic human genetics terminology (15) | 1.7 The role of behavioral, social and environmental factors to modify or influence genetics in manifestation of disease (3) |
| 1.2 Basic patterns of biological inheritance and variation (10) | 1.10 The potential physical and/or psychosocial benefits, limitations and risks of genetic information for individuals, family members and communities (7) |
| 1.3 How identification of disease-associated genetic variations facilitates development of prevention, diagnosis and treatment options (11) | 1.12 The resources available to assist clients seeking genetic information or services, including the types of genetics professionals available and their diverse responsibilities (2) |
| 1.4 The importance of family history in assessing predisposition to disease (8) | 1.13 The components of the genetic counseling process and the indications for referral to genetic specialists (14) |
| 1.5 The role of genetic factors in maintaining health and preventing disease (13) | 2.3 Ability to explain basic concepts of probability and disease susceptibility and the influence of genetic factors in maintenance of health and development of disease (7) |
| 1.6 The difference between clinical diagnosis of disease and identification of genetic predisposition to disease (9) | |
| 1.14 The indications for genetic testing and/or gene-based interventions (2) | |
| 2.1 Ability to gather genetic family history information, including an appropriate multigenerational family history (1) | |

*Certain assessment questions covered multiple competencies.

0.08. The genetic counseling content specific scores also increased in average absolute score from 69% (11/16) to 75% (12/16) but was not statistically significant with a p-value of 0.14.

Subjectively, trainers noted significant engagement but felt too much material was delivered over the course of the 3 days and too little time was available to review difficult concepts such as pedigree generation, inheritance patterns and all genetic counseling principles especially facilitated medical decision making.

Participants overall reported high levels of satisfaction with individual lectures and the overall program though on average participants felt the training was too short (**Table 2**). Multiple comments expressed satisfaction with the training and stated the importance of the delivered material for Ethiopians. The following is an example of one of these comments:

"*Very excellent, clear, and concise presentation with a positive and friendly vibe on a much-needed topic which has not been given emphasis till now.*"

An evaluation of the curriculum was performed by team members following analysis of pre- and post-assessment results, satisfaction scores and trainer observations. Though satisfaction scores were high for all delivered material, trainers noted too little time to adequately explain many aspects of the curriculum and especial difficulty conveying content focused on genetic counseling principles such as facilitated decision making. Given this it was decided to adjust the curriculum for Cohort 2 to increase the focus on medical knowledge material with future work to focus on genetic counseling principles.

Cohort 2 pre-workshop assessment showed a total score of 63% (19/30) which was identical to the baseline scores of Cohort 1. The post-workshop score increased to 73% (22/30) with the difference in scores achieving statistical significance (p = 0.043). The medical knowledge content specific scores increased in absolute score from 57% (8/14) to 79% (11/14) and this increase was statistically significant (p = 0.01). There was no change in scores for genetic counseling content material with a pre- and post-workshop score of 69% (11/16; p = 1.0). Importantly, while 15 participants were in attendance for all didactic aspects of the training, a number (n = 8) were unable to arrive in time for the pre-workshop assessment. As a result of this their post-workshop assessments were excluded from the above analysis. When their

**Table 2. Cohort satisfaction data.**

|  | Cohort 1 | Cohort 2 |
|---|---|---|
| The workshop was consistently clear, concise, and well-organized | 4.4 | 4.5 |
| The lecture material was useful in preparing me to deliver genetic counseling in the future | 4.5 | 4.6 |
| The small group exercises were effective in applying the delivered material | 4.3 | 4.4 |
| The role-playing exercises were effective in applying the delivered material | 4.4 | 4.5 |
| After completion of the workshop I feel better prepared to deliver genetic counseling in the future | 4.3 | 4.5 |
| I plan to use this training in the future | 4.5 | 4.6 |
| I would recommend this training to others | 4.6 | 4.8 |
| This workshop was effective at training providers to deliver condition-specific genetic counseling in the departments of Ob/Gyn and Pediatrics | 4.5 | 4.7 |
| The speakers were effective presenters | 4.5 | 4.7 |
| The use of an interpreter aided in my understanding of the material | 4.4 | 4.4 |
| Workshop length (1 = Much too short—5 = Much too long) | 2.3 | 2.8 |

Unless stated 1 = Strongly Disagree, 2 = Disagree, 3 = Neutral, 4 = Agree, 5 = Strongly Agree.

scores were included though there was no change in total post-assessment group score or scores of the medical knowledge or genetic counseling sections.

Cohort 2 participants also reported high levels of satisfaction with individual lectures and the overall program (**Table 2**). The following is an example of one comment from a trainee:

> "*It was good training program. I like it please keep it up. I would like to train other trainees in the future because it's interesting.*"

## Competency mapping

Of the core competencies necessary for all health professional, an understanding of genetics terminology and the components of genetic counseling were most represented in our assessments. **Fig 1** shows the comparison between Cohort 1 and Cohort 2 of all assessed competencies. No statistical difference was noted for Cohort 1 in any competencies (medical knowledge-specific or genetic counseling-specific) (**Fig 1A**). In Cohort 2 there were statistically significant increases in scores for the following domains: 1.1 Basic human genetics terminology (p = 0.0069), 1.2 The basic patterns of biological inheritance and variation, both within families and within populations (p = 0.01), 1.3 How identification of disease-associated genetic variation facilitates development of prevention, diagnosis and treatment options (p = 0.021), 1.4 The importance of family history in assessing predisposition to disease (p = 0.046), 1.5 The role of genetic factors in maintaining health and preventing disease (p = 0.013) and 1.6 The difference between a clinical diagnosis of disease and identification of a genetic predisposition to disease (p = 0.014) (**Fig 1B**). These competencies were all related to human genetics medical knowledge. No significant difference was seen for the remaining medical knowledge competencies or for any genetic counseling competencies.

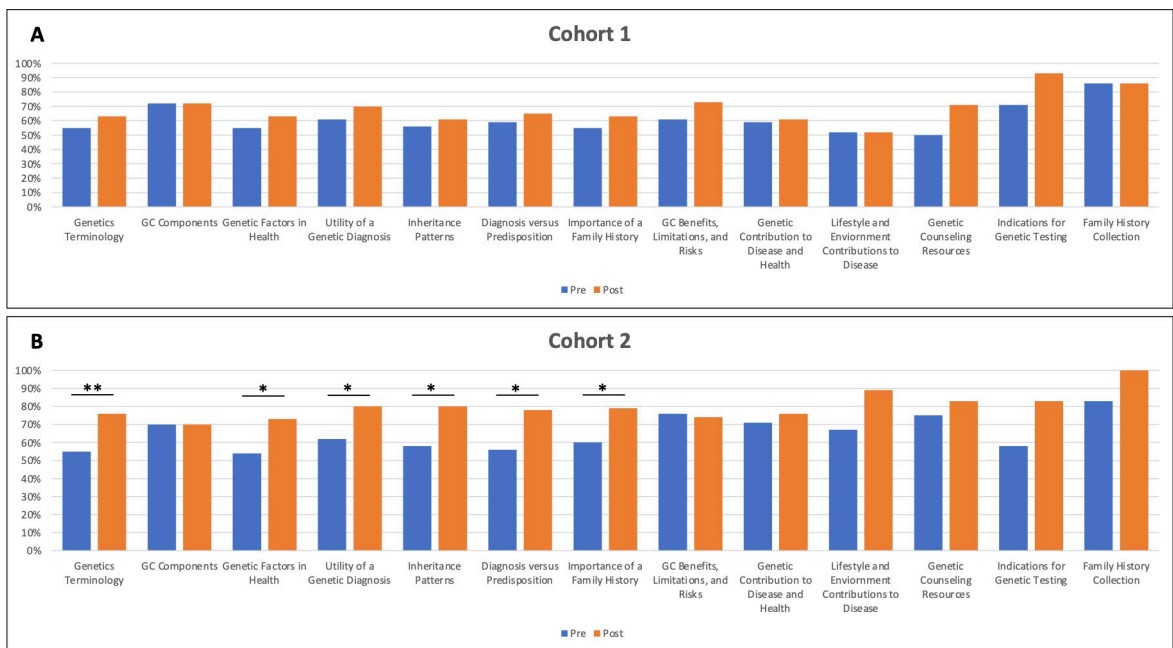

**Fig 1. Cohort 1 and Cohort 2 pre- and post-assessment scores by competency.** A: Cohort 1 scores showing no significant score differences. B: Cohort 2 scores. * = p-value 0.05–0.01; ** = p-value < 0.01.

## Barriers encountered

As expected, multiple barriers were encountered throughout the curriculum development, implementation and post-curriculum follow-up. These barriers were catalogued under the following categories: Logistical, Financial, Cultural and System-specific. While a number were anticipated and accounted for ahead of time the unanticipated barriers ultimately were the most difficult to navigate and had the most negative effect on implementation and follow-up. Logistical barriers were anticipated and included arranging trainee and trainer transportation to the training location, reserving a training location, securing audiovisual equipment and stationary ahead of time, printing out surveys and assessments, ensuring trainees had time excused for training and identifying trainees well ahead of the training start date. Anticipated financial barriers included paying trainer travel expenses, paying for hotel conference rooms needed for Cohort 2 training and providing snacks and lunches for trainees in both cohorts. A major unanticipated financial barrier encountered for Cohort 2 was the expectation of a per diem payment for trainees as the training was occurring off-campus. While this did not hinder engagement during the training it became an issue following training which may have contributed to difficulty with follow-up after the workshop. Overall, we noted difficulty identifying if nurses used their training either through the completion of case logbooks or through attempts at direct communication. Major anticipated cultural barriers included the curriculum being taught in English and the teaching of Western genetic counseling principles. Based on this, Cohort 2 training was altered. An unanticipated systems-specific barrier was the rotation of nurses to other areas of the hospital. Following Cohort 1 training it was identified several trained nurses were rotated off Pediatrics or Obstetrics to clinical locations where their training was of less utility. Prior to training of Cohort 2, SPHMMC agreed to consider nurse training prior to rotating them off Pediatrics or Obstetrics. A summary of barriers and considerations are listed in **Table 3**.

## Discussion

The rising impact of NCDs combined with increasing access to genetic testing will make the incorporation of genetic counseling services vital in LMICs. South Africa, which established its first genetic counseling training program in the 1980s, is the only country in SSA with a formalized genetic counseling training program [20]. In all of Africa, with a total population of 1.3 billion people, there is estimated to be less than 50 formally trained genetic counselors for the entire continent [12]. Given these variables and based on our experiences in Ethiopia, where genetic testing services were available well before providers felt comfortable providing genetic counseling, it is anticipated other low-income countries are likely to find themselves in need of genetic counseling services [17]. We designed and implemented the country's first broad genetic counseling training given the gap in the availability of genetic testing services in Ethiopia and an absence of formalized genetic counseling services. To our knowledge it is the first time a broad-based genetic counseling curriculum has been taught in a low-income country. An iterative process of curriculum development was used to train two cohorts of Ethiopian nurses. The final curriculum was highly focused on medical knowledge and utilized supplemental learning. The training and individual lectures were liked by trainees, met stated lecture objectives and were effective in increasing the medical knowledge of trained nurses.

Significant curriculum modifications were made following our experiences with the training of Cohort 1. The volume of delivered content combined with limited previous exposure to medical genetics resulted in significant difficulties with content retention. No statistically significant improvements were noted on post-assessment of this cohort. Additionally, as the content covering Western genetic counseling principles was difficult to grasp and much less likely

**Table 3. Barrier to global health curriculum implementation and barrier considerations.**

| Barriers | Considerations |
|---|---|
| **Logistical** | *Transportation and Lodging* |
| | • Special visa and/or immunization requirements |
| | • Telecommunication with collaborators upon arrival |
| | • In-country travel to and from various locations |
| | • Location of lodging relative to training location |
| | *Facilities* |
| | • Audiovisual equipment considerations such as region-specific electrical outlets and computer connections |
| | • Training location and alignment with planned teaching style (didactics, small group, etc) |
| | *Supplies* |
| | • Printing of training material prior to travel or upon arrival |
| **Financial** | *Facilities and Food* |
| | • Lunch and/or snacks for trainees |
| | • Potential facility usage fees |
| | *Transportation and Per diem* |
| | • Consideration of travel distance to training location for trainees and any associated bus or taxi fees |
| | • Need for a per diem for trainees especially if missing clinical duties for training |
| **Cultural** | *Language* |
| | • Necessity of a translator for trainees or translation of training materials |
| | *Content Appropriateness* |
| | • Alignment with local medical and/or cultural practices |
| | • Is training being primarily driven by local stakeholders? |
| **Systems-specific** | *Training Usability* |
| | • Is training being delivered to those most likely to use the learned material? |
| | • Trainee responsibilities following training and availability to utilize training |
| | • Will trainees be unnecessarily burdened with additional responsibilities following training? |

to be retained, especially if not deliberately practiced following training, it was decided to increase the focus on medical knowledge content. While this sacrifice limited the comprehensiveness of the curriculum, the increased medical knowledge focus was emphasized given previously shown difficulties with the training of communication skills which are necessary for the genetic counseling principles covered by the workshop [21]. Following these changes subjective and objective improvements in material comprehension and retention were noted. The increased time available for medical knowledge content allowed for additional problem-based examples to be used. This required the trainer and facilitators to identify in real time which concepts were difficult to grasp. For instance, the concept of children of a parent affected with an autosomal recessive condition being obligate carriers was noted to be difficult to grasp. When this was identified impromptu problem sets illustrating this were created and reviewed with the trainees.

With these modifications we noted significant improvements in medical knowledge retention in Cohort 2. To identify which program components were most effectively taught competency mapping unsurprisingly revealed medical knowledge-specific competency scores were significantly improved especially when compared to genetic counseling-specific competencies. The education of human genetics terminology was the most positively affected topic with inheritance patterns, the utility of a genetic diagnosis, the importance of family history, the role of genetics in health and disease prevention and the difference between a genetic diagnosis

and a genetic predisposition to disease also significantly improved. The assessment results were in line with the subjective observations of the trainers of Cohort 2 with an obvious improvement in trainee comprehension of the medical knowledge compared to experiences with Cohort 1. The effectiveness of the delivery of genetics medical knowledge is similar to that seen in previous genetic counseling training delivered in Kenya. This program, which focused on the genetics of retinoblastoma, utilized didactic sessions, small group discussions and role-playing and noted significant improvements in knowledge following training [8]. Similar to our work, this workshop was well liked by Kenyan trainees though poor retention of delivered material was noted one year later. It is also worth noting our assessment was not designed to provide a passing score but rather was utilized as a tool to assess curriculum effectiveness. A passing score for assessments, which can be determined by various methods including the Angoff method, could be useful for evaluations as genetic counseling expands into a formal profession in Ethiopia [22]. It is also important to note our initial intent was for our workshops to follow a Train the Trainer model. While we did incorporate two trainees from Cohort 1 as trainers for Cohort 2 we are uncertain of the ability or comfort of Cohort 2 participants to train others. The reasons for this are multifactorial and include many of the barriers outlined in Table 3 as well as the new and complex subject material and uncertainties regarding the minimum level of competency needed to function as an effective trainer. Additional work will be required to enable trained nurses to competently and comfortably function as future trainers.

While the training of nurses at SPHMMC was our primary goal we additionally catalogued the difficulties experienced to assist others performing or planning similar work. Multiple barriers were identified that warrant consideration by any group prior to similar work. Importantly, baseline suitability data was obtained showing both patients and providers were interested in medical genetics services in various forms [16, 17]. This needs assessment is an important aspect of the introduction of genetic services, especially if programs are deployed in partnership with high-income institutions [6, 13]. Each cohort trained resulted in the identification of new and unforeseen barriers to sustainable program implementation. The realization that some nurses were regularly rotated to different clinical areas based on clinical need resulted in decreased ability by the trained nurses to use the genetics knowledge and skills. While this was addressed for Cohort 2, we subsequently encountered unforeseen financial barriers that may have contributed to limited long term follow-up with the trainees. While satisfaction and assessment scores showed the training was successful in its stated objectives the practical use of the training was never fully actualized. The understanding and expectation of unforeseen barriers likely to arise during training development and deployment is important for others to anticipate. Barriers to project implementation do require groups to determine if the obstacles are simply the result of logistical difficulties or are symptoms of larger more problematic issues such as culturally inappropriate content or the many other ethical issues that may arise though global health research collaborations [13, 23]. The trainers identified the difficulty with teaching Western genetic counseling principles was likely the result of unfamiliarity with the material but also importantly limited alignment with the current practice patterns and established norms for provider-patient relationships in Ethiopia. The three topics removed for Cohort 2 were Facilitated Decision-making, Delivering Bad News and Handling Patient Emotion. Though these areas of practice are used regularly in Ethiopia, previous work has shown differences between Western genetic counseling practice and practices in Ethiopia surrounding patient-provider communication, patient autonomy and decision-making especially [24, 25]. For example, previous work in Ethiopia showed 40% of all medical decisions in rural Ethiopian patients were made by a woman's husband [26]. After reflecting on this

difference and Cohort 1's experience and feedback it was appropriately decided to remove this content from Cohort 2's training.

While we were unable to establish a regularly utilized cohort of trained nurses a number of important accomplishments were achieved that will further our group's efforts to introduce medical genetics in Ethiopia. These achievements include identifying patient and provider interest in genetic counseling services, further optimizing medical genetics educational practices in Ethiopia, identifying barriers to future capacity building projects, strengthening the relationship between our collaborating institutions and establishing a cohort of previously trained nurses optimal for involvement in future capacity building projects. Additional work will focus on identifying best practices for trained nurse education and deployment and how best to utilize virtual teaching given the travel implications that have resulted from the COVID-19 pandemic. Future work will also consider the early establishment of a SPHMMC oversight committee, responsible for the follow-up of delivered material, which may increase the likelihood of training utilization and follow-up.

In the future all countries in SSA are likely to gain access in some form to genetic testing services. With less than 50 formally trained genetic counselors for the entire continent it will be important to address how genetic counseling services are utilized by African healthcare providers. This work shows the feasibility of training nurses in the medical knowledge necessary to discuss various medical genetics topics with patients and families. Future work will need to consider how training and utilization of learned skills are best utilized and incorporated to ongoing practice patterns and how the incorporation of physicians into future training will augment program success. SSA countries, like Ethiopia, will need to consider how best to introduce genetic services by delicately balancing the field's benefits to patients, families and providers with the impact genetic services have on a country's social, cultural, religious and legal environment.

## Supporting information

**S1 Assessment. MiGene training program post-assessment.**
(DOCX)

**S1 Checklist. CIRHT genetic counselor checklist.**
(DOCX)

## Author Contributions

**Conceptualization:** Shane C. Quinonez, Atnafu Mekonnen Tekleab, Ayalew Marye, Delayehu Bekele.

**Data curation:** Bridget C. O'Connor, Michelle F. Jacobs.

**Formal analysis:** Erika Hanson.

**Funding acquisition:** Shane C. Quinonez.

**Investigation:** Shane C. Quinonez, Bridget C. O'Connor, Michelle F. Jacobs, Atnafu Mekonnen Tekleab, Ayalew Marye, Delayehu Bekele.

**Methodology:** Shane C. Quinonez, Bridget C. O'Connor, Michelle F. Jacobs, Atnafu Mekonnen Tekleab, Ayalew Marye, Delayehu Bekele, Beverly M. Yashar, Abate Yeshidinber.

**Project administration:** Atnafu Mekonnen Tekleab, Ayalew Marye, Delayehu Bekele.

**Resources:** Shane C. Quinonez, Bridget C. O'Connor, Michelle F. Jacobs, Atnafu Mekonnen Tekleab, Ayalew Marye, Delayehu Bekele, Abate Yeshidinber, Getahun Wedaje.

**Software:** Erika Hanson.

**Supervision:** Shane C. Quinonez.

**Validation:** Shane C. Quinonez, Bridget C. O'Connor, Michelle F. Jacobs, Erika Hanson.

**Visualization:** Shane C. Quinonez, Erika Hanson.

**Writing – review & editing:** Shane C. Quinonez, Bridget C. O'Connor, Michelle F. Jacobs, Atnafu Mekonnen Tekleab, Ayalew Marye, Delayehu Bekele, Beverly M. Yashar, Erika Hanson, Abate Yeshidinber, Getahun Wedaje.

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
