## [Decision Letter · Decision Letter 0]

26 May 2021

PONE-D-21-03085

The Introduction of Genetic Counseling in Ethiopia: Results of a Training Workshop and Lessons Learned

PLOS ONE

Dear Dr. Quinonez,

Thank you for submitting your manuscript to PLOS ONE. After careful consideration, we feel that it has merit but does not fully meet PLOS ONE’s publication criteria as it currently stands. Therefore, we invite you to submit a revised version of the manuscript that addresses the points raised during the review process.

We look forward to receiving your revised manuscript.

Kind regards,

Vijayaprakash Suppiah, PhD

Academic Editor

PLOS ONE

Journal Requirements:

Reviewers' comments:

Reviewer's Responses to Questions

**Comments to the Author**

1. Is the manuscript technically sound, and do the data support the conclusions?

Reviewer #1: Yes

Reviewer #2: Yes

2. Has the statistical analysis been performed appropriately and rigorously? 

Reviewer #1: Yes

Reviewer #2: Yes

3. Have the authors made all data underlying the findings in their manuscript fully available?

Reviewer #1: Yes

Reviewer #2: Yes

4. Is the manuscript presented in an intelligible fashion and written in standard English?

Reviewer #1: Yes

Reviewer #2: Yes

5. Review Comments to the Author

Reviewer #1: I think this work highlight really important topics: The increasing relevance of NCD also in low and middle-income countries, therole of genetic counseling in this field, the importance of a specific knowldege and habilities for professionals and the last but not the least real world barriers to implement this expertise in these countries.

I would just to note some minor comments:

1- Keep abbreviations throughout the text

2- It should interesting to know the interval of time the surveys (pre and post workshops) were completed in both cohorts

3- Although the statistical analyzes are ok, I think they could be more complete, for example, what cases Welch's correction was necessary.

4- Figure 1 is difficult to read and sholud be attached again

Reviewer #2: This is a timely manuscript as it is an important topic in equitable access to genetic information and genetic counselling to improve risk assessment, management and treatment of genetic diseases in low and medium income countries like the Sub-Saharan African countries.

1) In the first paragraph of page 3, when describing the medical genetics need in the Ethiopia, there is no information about the current services that are available for example, number of clinical genetics services, number of clinical geneticist and genetics counsellors (if any) and the population size. This would be useful to explain current practice in the country.

2) Line 138 : .. .."Ethiopian physician who translated confusing topics to trainee ": please mention what language was used.

3) Line 140 : " Attempts were made to choose nurses with adequate English fluency" - to understand possible language barriers, please describe the language(s) that is commonly used by the nurses in clinic. This was also mentioned as a barrier in line 274.

4) Line 146 / Line 147 and line 357 to 359 :".. decrease the content on covering Western-based genetic counselling principles." It was mentioned that these principles were the most difficult to grasp by nurses and were more likely to misalign with established Ethiopian cultural norms. Which principles are difficult and what is the established cultural norms that is is misaligned with? It is not clear as the Training Programme Assessment covered broadly on empathy, confidentiality and delivery of bad news. This was also mentioned as a barriers in line 274. Please explore this part about Western genetic counselling principles that are misaligned.

5) Under Results - Line 200 : "the genetic counselling content specific scores" : the questions in the assessment for genetic counselling content such as Q2 / Q4 / Q6 and others are True/False but followed by 'please explain'. Are there any scores given to the explanation or it is for collection of field notes?

6) The practical barriers were well described in the discussion section and some good suggestions were given moving forward. Line 377 : "This works shows the feasibility of training nurses in the medical knowledge necessary to discuss various medical genetics topics with patients and families.". The result has shown an improvement in medical genetics knowledge and good feedback from the nurses trained but the logistical and other barriers have made it not feasible to train nurses using this 'Train the Trainer' method? Please comment.

6. PLOS authors have the option to publish the peer review history of their article (what does this mean?). If published, this will include your full peer review and any attached files.

Reviewer #1: No

Reviewer #2: No

---

## [Author Response · Author response to Decision Letter 0]

25 Jun 2021

Response to Reviewers

Reviewer #1 Comments: 

Reviewer: Keep abbreviations throughout the text.

Response: Thank you for point this out. This has been addressed throughout the manuscript. 

Reviewer: It should interesting to know the interval of time the surveys (pre and post workshops) were completed in both cohorts

Response: We have added this information to the Materials and Methods section. We have specifically added the following text: 

“For Cohort 1 and Cohort 2 pre-assessments were completed on the first day of the workshop. Individual lecture evaluations were completed at the end of each day with the entire curriculum evaluation and post-assessment completed at the end of the final day of the workshop” 

Reviewer: Although the statistical analyzes are ok, I think they could be more complete, for example, what cases Welch's correction was necessary.

Response: We have revised this section of the Materials and Methods section. Upon further review of the analyses used in the paper we found Welch’s correction was not used for any of the data presented so this was removed. The section now reads as follows: 

“Assessment data and survey responses were collected in Excel. R was used to calculate pooled Student’s t-tests to assess statistically significant differences between pre- and post-assessments for each competency

Reviewer: Figure 1 is difficult to read and should be attached again. 

Response: This has been reuploaded as a higher resolution document. If still blurry the downloadable version should be clearer. Thank you for pointing this out. 

Reviewer #2 Comments: 

Reviewer: In the first paragraph of page 3, when describing the medical genetics need in Ethiopia, there is no information about the current services that are available for example, number of clinical genetics services, number of clinical geneticist and genetics counsellors (if any) and the population size. This would be useful to explain current practice in the country.

Response: This information has been added. We have added the following text: 

“Currently there are no established medical genetics services with no in-country clinical geneticists or genetic counselors.

Reviewer: Line 138 : .. .."Ethiopian physician who translated confusing topics to trainee ": please mention what language was used.

Response: This information was added. Specifically, the following text is now in the Materials and Methods section: 

“Since the Ethiopian medical curriculum is taught in English, the curriculum was delivered in English by three study authors (SQ, BO, MJ) who were aided by an Ethiopian physician who translated confusing topics to trainees and/or questions to the trainers in Amharic, the principal language in Ethiopia. While medical education in Ethiopia is generally taught in English, Amharic and Afan Oromo are the most commonly spoken languages at SPHMMC with patients.”

Reviewer: Line 140 : " Attempts were made to choose nurses with adequate English fluency" - to understand possible language barriers, please describe the language(s) that is commonly used by the nurses in clinic. This was also mentioned as a barrier in line 274.

Response: In addition to the text shown above additional information was added. The full section includes the following text: 

“Since the Ethiopian medical curriculum is taught in English, the curriculum was delivered in English by three study authors (SQ, BO, MJ) who were aided by an Ethiopian physician who translated confusing topics to trainees and/or questions to the trainers in Amharic, the principal language in Ethiopia. While medical education in Ethiopia is generally taught in English, Amharic and Afan Oromo are the most commonly spoken languages at SPHMMC with patients. Attempts were made to choose nurses with adequate English fluency for ease of training.”

Reviewer: Line 146 / Line 147 and line 357 to 359 :".. decrease the content on covering Western-based genetic counselling principles." It was mentioned that these principles were the most difficult to grasp by nurses and were more likely to misalign with established Ethiopian cultural norms. Which principles are difficult and what is the established cultural norms that is is misaligned with? It is not clear as the Training Programme Assessment covered broadly on empathy, confidentiality and delivery of bad news. This was also mentioned as a barriers in line 274. Please explore this part about Western genetic counselling principles that are misaligned.

Response: This is an important point. To address this, we included the following text which included additional references now in the manuscript: 

“The three topics removed for Cohort 2 were Facilitated Decision-making, Delivering Bad News and Handling Patient Emotion. Though these areas of practice are used regularly in Ethiopia, previous work has shown differences between Western genetic counseling practice and practices in Ethiopia surrounding patient-provider communication, patient autonomy and decision-making especially [24,25]. For example, previous work in Ethiopia showed 40% of all medical decisions in rural Ethiopian patients were made by a woman’s husband [26]. “ 

Reviewer: Under Results - Line 200 : "the genetic counselling content specific scores" : the questions in the assessment for genetic counselling content such as Q2 / Q4 / Q6 and others are True/False but followed by 'please explain'. Are there any scores given to the explanation or it is for collection of field notes?

Response: This information was added to the Materials and Methods section. The following 

text was added: 

“For True/False assessment questions though individuals were able to provide an 

explanation for their True/False response this aspect of the question was not scored.”

Reviewer: The practical barriers were well described in the discussion section and some good suggestions were given moving forward. Line 377 : "This works shows the feasibility of training nurses in the medical knowledge necessary to discuss various medical genetics topics with patients and families.". The result has shown an improvement in medical genetics knowledge and good feedback from the nurses trained but the logistical and other barriers have made it not feasible to train nurses using this 'Train the Trainer' method? Please comment.

Response: This important point was addressed with the following text added to the Discussion: 

“It is also important to note our initial intent was for our workshops to follow a Train the Trainer model. While we did incorporate two trainees from Cohort 1 as trainers for Cohort 2 we are uncertain of the ability or comfort of Cohort 2 participants to train others. The reasons for this are multifactorial and include many of the barriers outlined in Table 3 as well as the new and complex subject material and uncertainties regarding the minimum level of competency needed to function as an effective trainer. Additional work will be required to enable trained nurses to competently and comfortably function as future trainers.”

---

## [Decision Letter · Decision Letter 1]

14 Jul 2021

The Introduction of Genetic Counseling in Ethiopia: Results of a Training Workshop and Lessons Learned

PONE-D-21-03085R1

Dear Dr. Quinonez,

We’re pleased to inform you that your manuscript has been judged scientifically suitable for publication and will be formally accepted for publication once it meets all outstanding technical requirements.

Kind regards,

Vijayaprakash Suppiah, PhD

Academic Editor

PLOS ONE

Reviewers' comments:

Reviewer's Responses to Questions

**Comments to the Author**

1. If the authors have adequately addressed your comments raised in a previous round of review and you feel that this manuscript is now acceptable for publication, you may indicate that here to bypass the “Comments to the Author” section, enter your conflict of interest statement in the “Confidential to Editor” section, and submit your "Accept" recommendation.

Reviewer #1: All comments have been addressed

Reviewer #2: All comments have been addressed

2. Is the manuscript technically sound, and do the data support the conclusions?

Reviewer #1: Yes

Reviewer #2: (No Response)

3. Has the statistical analysis been performed appropriately and rigorously? 

Reviewer #1: Yes

Reviewer #2: (No Response)

4. Have the authors made all data underlying the findings in their manuscript fully available?

Reviewer #1: Yes

Reviewer #2: (No Response)

5. Is the manuscript presented in an intelligible fashion and written in standard English?

Reviewer #1: Yes

Reviewer #2: (No Response)

6. Review Comments to the Author

Reviewer #1: Authors have adequately addressed my comments. I think it is notorious the effort of the authors to approach to the reality of genetic counseling in Ethiopia

Reviewer #2: (No Response)

7. PLOS authors have the option to publish the peer review history of their article (what does this mean?). If published, this will include your full peer review and any attached files.

Reviewer #1: No

Reviewer #2: No

---

## [Editor Report · Acceptance letter]

16 Jul 2021

PONE-D-21-03085R1 

The Introduction of Genetic Counseling in Ethiopia: Results of a Training Workshop and Lessons Learned 

Dear Dr. Quinonez:

I'm pleased to inform you that your manuscript has been deemed suitable for publication in PLOS ONE. Congratulations! Your manuscript is now with our production department. 

Kind regards, 

on behalf of

Dr. Vijayaprakash Suppiah 

Academic Editor

PLOS ONE